# Comparative Fluorescence In Situ Hybridization (FISH) Mapping of Twenty-Three Endogenous Jaagsiekte Sheep Retrovirus (enJSRVs) in Sheep (*Ovis aries*) and River Buffalo (*Bubalus bubalis*) Chromosomes

**DOI:** 10.3390/ani12202834

**Published:** 2022-10-19

**Authors:** Angela Perucatti, Alessandra Iannuzzi, Alessia Armezzani, Massimo Palmarini, Leopoldo Iannuzzi

**Affiliations:** 1National Research Council (CNR), Institute of Animal Production System on Mediterranean Environment (ISPAAM), Piazzale E. Fermi, 1, 80055 Portici, Italy; 2MRC-University of Glasgow Centre for Virus Research, 464 Bearsden Road, Glasgow G61-1QH, UK

**Keywords:** sheep, river buffalo, endogenous retrovirus, FISH-mapping, cytogenetic map

## Abstract

**Simple Summary:**

The genome of domestic sheep (*Ovis aries*) harbors at least twenty-seven copies of enJSRVs, endogenous retroviruses (ERVs) highly related to the exogenous and pathogenic Jaagsiekte sheep betaretrovirus (JSRV). Interestingly, some of these loci are insertionally polymorphic, that is they are present only in some individuals or populations of their host species. This differential distribution of enJSRVs has provided important insights into tracing host and viral evolution. In this study, we report the first comparative fluorescent in situ hybridization (FISH) mapping of genetically characterized enJSRVs in domestic sheep (2n = 54) and river buffalo (*Bubalus bubalis*, 2n = 50), and reveal a high conservation of enJSRVs chromosome localization between these two species.

**Abstract:**

Endogenous retroviruses (ERVs) are the remnants of ancient infections of host germline cells, thus representing key tools to study host and viral evolution. Homologous ERV sequences often map at the same genomic locus of different species, indicating that retroviral integration occurred in the genomes of the common ancestors of those species. The genome of domestic sheep (*Ovis aries*) harbors at least twenty-seven copies of ERVs related to the exogenous and pathogenic Jaagsiekte sheep retrovirus (JSRVs), thus referred to as enJSRVs. Some of these loci are unequally distributed between breeds and individuals of the host species due to polymorphic insertions, thereby representing invaluable tools to trace the evolutionary dynamics of virus populations within their hosts. In this study, we extend the cytogenetic physical maps of sheep and river buffalo by performing fluorescent in situ hybridization (FISH) mapping of twenty-three genetically characterized enJSRVs. Additionally, we report the first comparative FISH mapping of enJSRVs in domestic sheep (2n = 54) and river buffalo (*Bubalus bubalis*, 2n = 50). Finally, we demonstrate that enJSRV loci are conserved in the homologous chromosomes and chromosome bands of both species. Altogether, our results support the hypothesis that enJSRVs were present in the genomes of both species before they differentiated within the *Bovidae* family.

## 1. Introduction

Retroviruses possess the unique ability to integrate into the genome of infected cells. Occasionally, they can infect germline cells and give rise to endogenous retroviruses (ERVs), retroviral sequences transmitted vertically, in a Mendelian fashion, as part of the host genome. As such, ERVs represent fascinating tools to study both virus and host genome evolution [1]. ERVs have been found in all vertebrates studied to date, including fish, amphibians, birds, reptiles, and mammals [2]. Comparative genomic studies have shown that related species often share ERV families or specific ERV loci, and that, in many cases, homologous ERV sequences map at the same genomic locus in multiple species’ genomes, indicating that retroviral integration events occurred in the genomes of the common ancestors of those species [3,4].

In 2013, Garcia-Etxebarria and Jugo traced the evolutionary history of bovine ERVs (BoERVs) by performing computational analyses on the genomes of several bovid species, including cattle, sheep, goat, and water buffalo [5]. Interestingly, they found twenty-six BoERV families in all the species studied, suggesting that most of these families could be present in all members of the *Bovidae* family. However, they could not detect four BoERV families (i.e., BoERV24, BoERV26, BoERV28, and BoERV29) in sheep or goat genomes, indicating that these families may be specific to the *Bovinae* subfamily. The authors hypothesize that the majority of the BoERV families invaded the genome of the common ancestor of the *Bovidae* family approximately 20 million years ago (MYA) and only later, between 12MYA and 20MYA, the ancestors of the BoERV24, BoERV26, BoERV28, and BoERV29 families might have been inserted into the genome of the *Bovinae* subfamily ancestor [5].

The domestic sheep (*Ovis aries*) harbors at least twenty-seven ERV loci related to the exogenous and pathogenic Jaagsiekte sheep retrovirus (JSRV), thus referred to as enJSRVs [6,7]. Interestingly, some enJSRV loci are insertionally polymorphic, that is they are present only in some individuals or populations of their host species. This differential distribution has provided important insights into tracing the evolutionary dynamics of virus populations within their hosts [8].

Along these lines, in previous studies, we used enJSRVs to (i) investigate the history of sheep domestication [9], (ii) explore the molecular mechanisms through which the most recent enJSRV—enJSRV-26—eludes the restriction exerted by enJS56A1 (which entered the sheep genome before and during the speciation within the genus *Ovis*) [10], or (iii) study the chromosome location of some enJSRV loci by fluorescent in situ hybridization (FISH) analyses of metaphase R-banded sheep chromosomes [9,10].

In the present work, we extend the cytogenetic physical maps of both sheep and river buffalo chromosomes by FISH mapping fifteen and twenty-three additional enJSRVs, respectively. In addition, we perform the first comparative FISH mapping of genetically characterized enJSRVs in domestic sheep and river buffalo (*Bubalus bubalis*).

## 2. Materials and Methods

*Cell cultures.* Peripheral blood lymphocytes of sheep (*Ovis aries*, OAR; 2n = 54) (four animals) and river buffalo (*Bubalus bubalis*, BBU; 2n = 50) (three animals) were cultured for 72 h in Roswell Park Memorial Institute (RPMI)-1640 culture medium enriched with 15% fetal calf serum (FCS), concanavalin A (15 µg/mL), penicillin/streptomycin (0.1 mL), L-glutamine (0.1 mL), and one drop of sterile sodium heparin to prevent coagulation. In order to obtain R-banding patterns and cause metaphase arrest, 7 h prior to harvesting, cells were labeled with BrdU (15 μg/mL) and Hoechst 33258 (30 μg/mL) and, 6 h later, they were treated with colcemid (0.1 μg/mL). After incubation in a hypotonic solution (KCl 0.075 M at 37.5 °C for 20 min), cells were fixed three times with 3:1 methanol–acetic acid (*v*/*v*) solution. Cell suspensions were then spread onto a slide, and stored at −20 °C.

*Probes and FISH mapping.* FISH analysis was performed using standard procedures [11,12]. Briefly, cells were pre-treated overnight at 50 °C, and subsequently stained with Hoechst 33258 (25 μg/mL) for 10 min. Slides were then exposed to UV light for 20 min, washed with distilled water, and air-dried. Hybridization, chromosome staining, signal detection and image processing were carried out as already described [11,12]. Slides were mounted in antifade mounting medium with propidium iodide to visualize both FITC-signals and RBPI-banding using two microscope filter combinations. Two images for each metaphase were acquired with both FITC signals and RBPI-banding. Next, FITC signals were superimposed on RBPI-banding to get a precise position of FITC-signals on chromosome bands. Thirty metaphases for each probe were examined. Chromosome identification and banding numbering system followed the standard nomenclature of both species [13,14]. The bacterial artificial chromosome (BAC) clones used for sheep cytogenetic mapping have been already characterized and described elsewhere [6]. Notably, each BAC clone used to obtain the sequences of the various enJSRV loci was subjected to Southern blot analyses to verify that it contained only one locus. The list of sheep BAC clones of the CHORI-243 library containing the 27 enJSRVs used in this study is reported in Table 1:

## 3. Results

In the present study, we conducted FISH mapping on sheep and river buffalo chromosomes (or chromosome arms) using twenty-seven sheep BAC clones (Table 1). As reported in Table 2, we obtained good hybridization signals only with twenty-three BAC clones, allowing us to map the corresponding twenty-three enJSRV loci. Interestingly, we localized these loci on twelve different chromosomes (or chromosome arms) of both species. As reported in Table 1, we found that, for all the probes used, hybridization signals and chromosome bands localized in the same homologous chromosome pairs of both sheep and river buffalo (Table 2). For example, we found that the enJSRV-1 and enJSRV-10 loci map onto homologous sheep and river buffalo R-banded chromosomes, as shown in Figure 1. Moreover, the BAC containing the enJSRV-1 map onto two different chromosomes in both sheep (chromosomes 6 and 18) and river buffalo (chromosomes 7 and 20) (Figure 1 and Table 2). In addition, the enJSRV-15, enJSRV-20, enJSRV-21, and enJSRV-27 display the same chromosome localization as the enJSRV-1 in both species (OAR6q13 and BBU7q21, respectively) (Table 2). Finally, the enJSRV-2 and enJSRV-6 map very close in both sheep (OAR1q45 and OARiq43, respectively) and river buffalo homologous chromosomes (BBU1q45 and BBU1q43, respectively). Notably, only the enJSRV-7 maps onto the centromeric regions of all autosomes [10], probably due to the presence of highly repetitive sequences in the BAC clones, the centromeric regions being highly rich in these sequences.

## 4. Discussion

FISH represents a very powerful cytogenetic technique for mapping a particular genomic sequence on a chromosome [15,16], and better anchoring of radiation hybrid (RH) and genomic maps to specific chromosome regions [17,18,19,20]. In more recent years, FISH has also been recognized as a reliable diagnostic and discovery tool to evaluate genetic anomalies, by studying chromosomal aberrations in both metaphase and interphase nuclei [reviewed in [21,22,23,24,25], and defects in chromosome segregation during meiosis [26,27]. In addition, implementations of FISH with whole-chromosome painting have led to the generation of detailed comparative maps to study chromosomal homologies and divergences between related and unrelated species [28,29,30,31,32]. Finally, FISH has also become instrumental in generating detailed comparative maps to study gene order, conserved chromosomal regions, and chromosomal rearrangements between related and unrelated species [33,34,35,36].

Along these lines, FISH analyses unveiled the phylogenetic relationships between the *Caprinae* subfamily and the earliest-diverging *Bovinae* subfamily, by showing two main chromosome events occurring at the autosomes 9 and 14, and the sex chromosomes (mainly the X-chromosome). More specifically, in previous studies carried out in our group, we demonstrated that a chromosome transposition has occurred from the proximal region of *Bovinae* chromosome 9 to the proximal region of *Caprinae* chromosome 14, and that at least four chromosome rearrangements (i.e., three transpositions and one inversion) differentiated the *Caprinae* from the *Bovinae* X-chromosomes reviewed in [37].

A comparative FISH mapping of enJSRVs has been reported previously in sheep and goat cell lines [38]. In this study, the authors only partially found enJSRVs localized on the same homologous chromosome band of the two species. In addition, they found enJSRV loci in seven and eight chromosomes of sheep and goats, respectively.

In the present work, we report the first comparative FISH mapping between two species belonging to the *Caprinae* (*Ovis aries*) and *Bovinae* (*Bubalus bubalis*) subfamilies of the *Bovidae* family by using well-identified and genetically characterized enJSRVs [5,6]. We show that hybridization signals of enJSRVs are found in at least twelve different chromosomes (or chromosome arms) of both species, and that all mapped loci are conserved in homologous chromosome regions and chromosome bands of these two species (Table 1). However, since BACs contain large genomic inserts, it is entirely possible that, besides enJSRVs, we also simultaneously mapped other genes and sequences present on such clones that share homology with some river buffalo chromosomal regions. Indeed, since some BoERV families (i.e., BoERV24, BoERV26, BoERV28, and BoERV29) are specific to the *Bovinae* family [5], the same could have occurred for some enJSRV which are present in the *Caprinae* subfamily (i.e., *Ovis aries*) but not in the *Bovinae* subfamily (i.e., *Bubabus bubalis*).

We observed that, in both species, the BAC clones containing enJSRV-1 map onto two different chromosomal locations (Table 2). Interestingly, we found that enJSRV-7 is the only locus mapping at the centromeric regions of both sheep [10] and buffalo chromosomes.

The same results were achieved by FISH mapping several ERV in sheep chromosomes which exhibited abundant centromeric to the dispersed distribution of various endoviruses, probably due to the abundance of genomic organization ERV-related repetitive elements which are particularly present at the centromeric regions of the chromosomes [39].

Our comparative FISH mapping in two different bovid species further confirms the high degree of chromosome (and chromosome arm) conservation among bovids reviewed in [37]. In addition, our study supports the hypothesis that enJSRVs were present in the genomes of their bovine ancestor before the differentiation of the *Caprinae* subfamily (including *Ovis aries*) from the most ancient *Bovinae* subfamily (including *Bubalus bubalis*) [37,40]. These results are in agreement with those published in a previous study [5] tracing BoERVs evolution in several species of the *Bovidae* family, including cattle, sheep, goats, and water buffalo [5]. Interestingly, these authors found that most of the BoERV families are present in all the species studied, supporting the hypothesis that BoERVs entered the genome of the common ancestor of the *Bovidae* family about 20 MYA or less. In addition, they detected higher BoERV copy numbers in cattle compared to other bovid species, suggesting that an additional expansion of retroviral copies might have occurred in the cattle genome [5].

Interestingly, we found five enJSRV loci (enJSRV-1, enJSRV-15, enJSRV-20, enJSRV-21, and enJSRV-27) on the same chromosome band (OAR6q13/BBU7q21) (Table 2). Previous FISH mapping conducted in our group revealed two type-one loci on this very same chromosome band: the pyroglutamylated RFamide peptide receptor (QRFPR) and the translocation-associated membrane protein 1-like 1 (TRAM1L1) [41]. The RFamide peptide family consists of several groups, including the neuropeptide FF group, the prolactin-releasing peptide group, the gonadotropin inhibitory hormone group, the kisspeptin group, and the pyroglutamylated RFamide peptide (26RFa/QRFP) group [42]. Interestingly, pyroglutamylated RFamide peptide 43 has been proven to be a putative modulator of testicular steroidogenesis, playing an important role in reproduction [43]. Notably, ERVs are key in placental morphogenesis and mammalian reproduction [44]. TRAM1L1 seems to be closely related to chronic widespread musculoskeletal pain (CWP), a common disorder affecting about 5–15% of the population, and one of the main symptoms of fibromyalgia, which has been shown to be associated with an altered gut microbiome [45]. By using the sheep genome reference sequence (https://www.ncbi.nlm.nih.gov/genome/gdv/browser/genome/?id=GCF_016772045.1, accessed on 12 May 2018), we identified sixteen genes included between QRFPR and TRAM1L (Appendix A). In this table we reported some of the functions of those genes (including QRFPR and TRAM1L), mostly involved in anti-tumor immune response probably to counteract the presence of several enJSRVs in these chromosomic regions. Indeed, genomic amplification within the 6q13 region was detected, and it was found that the number of enJSRV-6q13 is correlated to the number of protective mutations [46].

## 5. Conclusions

To our knowledge, this is the first comparative FISH mapping of sheep and river buffalo chromosomes using genetically characterized enJSRVs. Interestingly, our results reveal a high degree of conservation of enJSRVs localization in the homologous chromosomes and chromosome bands of both species. These findings support the hypothesis that enJSRVs entered the host genome before the differentiation of the *Caprinae* subfamily from the earliest-diverging *Bovinae* subfamily of the *Bovidae* family. Finally, the present study extends the current genetic physical maps of sheep and river buffalo by mapping, respectively, fifteen and twenty-three additional enJSRV loci on the chromosomes and chromosome arms of these two species.

## Figures and Tables

**Figure 1 animals-12-02834-f001:**
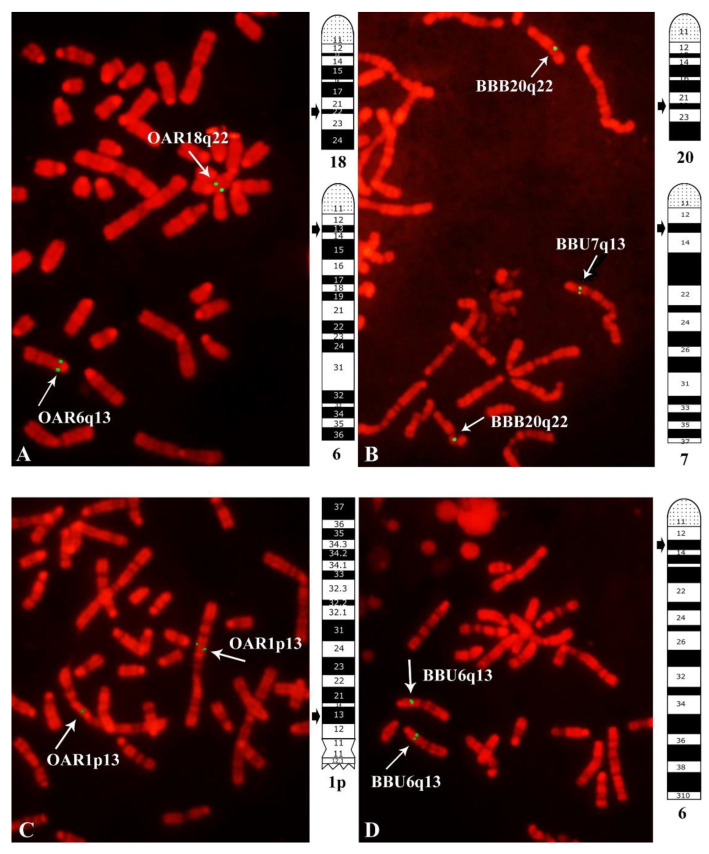
FISH mapping of enJSRV-1 (**A**,**B**) and enJSRV-10 (**C**,**D**) in sheep (OAR) and river buffalo (BBU) chromosomes. Two different images were taken, with hybridization FITC signal and with RBPI-banding. Subsequently, hybridization FITC signals were superimposed on RBPI-banding to get a precise localization of mapped loci in both species. Note that enJSRV-1 maps onto two different chromosomes in both species (**A**,**B**).

**Table 1 animals-12-02834-t001:** List of sheep BAC clones containing the 27 enJSRVs used in the present study.

Provirus	BAC Clone
enJSRV-1	6g20
enJSRV-2	6j6
enJSRV-3	7i2
enJSRV-4	15c2
enJSRV-5	19l17
enJSRV-6	33k3
enJSRV-7	35o24
enJSRV-8	36d13
enJSRV-9	36p8
enJSRV-10	42k15
enJSRV-11	44o16
enJSRV-12	45l23
enJSRV-13	48b6
enJSRV-14	52d13
enJSRV-15	57m3
enJSRV-16	57m4
enJSRV-17	65c8
enJSRV-18	67f9
enJSRV-19	70l9
enJSRV-20	81j8
enJSRV-21	84d10
enJSRV-22	88g3
enJSRV-23	90k11
enJSRV-24	94c2
enJSRV-25	98k22
enJSRV-26	102b15
enJSRV-27	14c2

**Table 2 animals-12-02834-t002:** Chromosomal localization of 23 enJSRVs in sheep (OAR) and river buffalo (BBU) chromosomes by FISH-mapping.

enJSRV	OAR	BBU
enJSRV-1	18q22	20q22
	6q13	7q13
enJSRV-2	1q45	1q45
enJSRV-3	14q24dist	18q24dist
enJSRV-4	2p21prox	3q21prox
enJSRV-6	1q41	1q41
enJSRV-7	centromeric	centromeric
enJSRV-8	3q21	4q21
enJSRV-10	14q24	18q24
enJSRV-11	1p13	6q13
enJSRV-12	19q24	21q24
enJSRV-13	14q24dist	18q24
enJSRV-14	3p24	12q34
enJSRV-15	6q13	7q21
enJSRV-16	10q24	13q24
enJSRV-17	19q24	21q24
enJSRV-18	11q17	3p22
enJSRV-19	1p23	6q25
enJSRV-20	6q13	7q13
enJSRV-21	6q13	7q13
enJSRV-22	15q23	16q23dist
enJSRV-24	18q24	20q24
enJSRV-26	2p25dist	3q25
enJSRV-27	6q13	7q13

## Data Availability

The data that support the findings of this study are available from the corresponding author, [AI], upon reasonable request.

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
