# Peer review of "Comparative Fluorescence In Situ Hybridization (FISH) Mapping of Twenty-Three Endogenous Jaagsiekte Sheep Retrovirus (enJSRVs) in Sheep (Ovis aries) and River Buffalo (Bubalus bubalis) Chromosomes"

_animals, 2022, doi:10.3390/ani12202834_

Round 1

Reviewer 1 Report

In this study the authors reported the first comparative fluorescent in situ hybridization (FISH) mapping of genetically characterized enJSRVs in domestic sheep (2n=54) and river buffalo (Bubalus bubalis, 2n=50). The manuscript is interesting, and the study falls into the scope of Animals, thus it could be published. Before this can be done, a major revision should be made as specified below.

1)    The introduction must be improved. The authors can explore what we know about the role of enJSRVs in the chromosome evolution. Are there similar studies in other animals group?

2)    Lines 56-58: What was the insights from the mapping of these eight enJSRV loci?

3)    Material and methods: How many individuals were used in the study? Considering that previous studies revealed that some enJSRV loci are insertionally polymorphic (lines 50-52), the FISH mapping must be performed on more than one individual per species.

4)    Figure 1: The complete metaphase should be presented;

5)    Lines 160-163. Please, add the divergence time;

6)    Lines 177-179. Please, move to the conclusion section;

7)    Similar to the introduction, the discussion must be improved, mainly about the role of enJSRVs in the chromosome evolution. Are there similar studies in other animals group to compare? In fact, I found two papers in Sheep that the authors may find interesting:

Mustafa, S.I.; Schwarzacher, T.; Heslop-Harrison, J.S. The Nature and Chromosomal Landscape of Endogenous Retroviruses (ERVs) Integrated in the Sheep Nuclear Genome. DNA 2022, 2, 86-103. https://doi.org/10.3390/dna2010007

Cumer, T., Pompanon, F. & Boyer, F. Old origin of a protective endogenous retrovirus (enJSRV) in the Ovis genus. Heredity 122, 187–194 (2019). https://doi.org/10.1038/s41437-018-0112-z

Author Response

Comments and Suggestions for Authors

In this study the authors reported the first comparative fluorescent in situ hybridization (FISH) mapping of genetically characterized enJSRVs in domestic sheep (2n=54) and river buffalo (Bubalus bubalis, 2n=50). The manuscript is interesting, and the study falls into the scope of Animals, thus it could be published. Before this can be done, a major revision should be made as specified below.

1)The introduction must be improved. The authors can explore what we know about the role of enJSRVs in the chromosome evolution. Are there similar studies in other animals group?

- Done

2)Lines 56-58: What was the insights from the mapping of these eight enJSRV loci?

- This part has been better rearranged now.

3)Material and methods: How many individuals were used in the study? Considering that previous studies revealed that some enJSRV loci are insertionally polymorphic (lines 50-52), the FISH mapping must be performed on more than one individual per species.

- Done

4)Figure 1: The complete metaphase should be presented;

- Done

5)Lines 160-163. Please, add the divergence time;

- Done (we refer to a previous study reporting these data)

6)Lines 177-179. Please, move to the conclusion section;

- Done

7) Similar to the introduction, the discussion must be improved, mainly about the role of enJSRVs in the chromosome evolution. Are there similar studies in other animals group to compare? In fact, I found two papers in Sheep that the authors may find interesting:

- Done

Mustafa, S.I.; Schwarzacher, T.; Heslop-Harrison, J.S. The Nature and Chromosomal Landscape of Endogenous Retroviruses (ERVs) Integrated in the Sheep Nuclear Genome. DNA 2022, 2, 86-103. https://doi.org/10.3390/dna2010007

Cumer, T., Pompanon, F. & Boyer, F. Old origin of a protective endogenous retrovirus (enJSRV) in the Ovis genus. Heredity 122, 187–194 (2019). https://doi.org/10.1038/s41437-018-0112-z

Both papers have been cited now

We wish this revised version will find your agreement.

Best regards

Dr. Alessandra Iannuzzi

Reviewer 2 Report

To authors,

The manuscript describes the mapping of the BACs that supposedly contain retroviral sequences of Jaagsiekte sheep retrovirus (JSRV). Authors found that BACs are localized on homologous chromosome regions in sheep and buffalo. For one BAC there were two locations in the genome. One of the genomic regions had multiple BAC’s localisation. One BAC was localised in the centromere.

It should be indicated whether the authors conducted a PCR experiment to verify whether each BAC clone has an insertion of the enJSRV before conducting FISH.

I have some doubt whether calling mapping of the BACs, containing enJSRV is the same as saying mapping enJSRV, unless it was for a fact shown that these BACs have an insert of the enJSRV AND also shown that genomes of both sheep and buffalo contain enJSRVs. I assume that the authors have purchased the same BACs as in the original paper that has identified 27 enJSRV in the sheep genome. The names of the BACs in the CHORI library have to be listed in the table. However, it happens sometimes that some BACs may not contain the intended sequence. So if authors want to claim that they have mapped not only BACs but BACs containing enJSRV or even claim the mapping of enJSRVs themselves, then we readers have to be sure that enJSRVs are indeed part of these BACs. Otherwise, I would not be calling it mapping of the enJSRV. 

Mapping enJSRV itself would include the direct amplification of the enJSRV from a sheep genome, labeling the PCR product, and mapping it onto sheep and buffalo chromosomes. The FISH experiment, of course, would be much trickier in such a case as the probe would be much smaller than BAC. 

Figure 1. Suggestion - at the bottom left corner of each FISH image put the name of the mapped retrovirus (in the white font?). At the bottom of the image put OAR and BBB under respective ideograms.

The authors say in the discussion that they have previously mapped two type 1 loci into one of the bands that also contains several insertions of enJSRVs. Well, this band has much more genes as shown in Gene Map Viewer or USCIS Genome Browser: for example, genes PRDM5, SYNP02, NDST3, and many more that are situated between QRFPR and TRAM1L1 genes described in the manuscript. If retrovirus accumulation in this band indeed affects the evolution of the mammalian genome then it makes sense to look at GO functions of ALL the genes in this particular band, not only on the 2 genes mapped by FISH. And then make a conclusion on whether enJSRV inserts may have somehow affected nearby genes.

It would be interesting to extend this study beyond Bovidae and to find a point on a phylogenetic tree of Cetartiodactyla where the genome integration of the retrovirus has occurred. This can at first be done bioinformatically by mining the cetartiodactyl genomes beyond Bovidae for enJSRVs.

However, first, we need to be sure that authors are not simply mapping ovine BACs in sheep and goat (those of course would localize on homologous chromosomes in these species!), but mapping BACs, containing enJSRV. Also, we need to be sure that when we are localizing some BAC in buffalo and claiming that the BAC has enJSRV, we are not simply mapping all sequences other than enJSRVs that are present in the BAC but indeed mapping enJSRVs itself. 

Some small suggestions are in the pdf text: provide hypotonic treatments details, add the word retrovirus into the title and etc.

Maybe Carlson et al., 2003 did not find homologous sequences in goat for many of the identified enJSRVs in sheep because of the divergence of enJSRVs. This is one potential indication that authors could be not mapping enJSRVs in buffalo but instead simply mapping BACs in buffalo. Meaning, that BAC can still map onto the buffalo chromosomes even if the buffalo genome does not contain enJSRV. 

Also, we need to be sure that the localization of the BAC on two chromosomes (enJSRV-1) is due to the genome containing multiple copies of the enJSRV and not multiple copies of some other genome fragment that is also present in the BAC. All this should either be discussed in the paper or a new probe that includes only enJSRV (produced by cloning the enJSRV into the plasmid or by PCR amplification) should be made and mapped (which would constitute a significant effort: the same or even bigger than the work presented in the manuscript).

Author Response

Comments and Suggestions for Authors

To authors,

The manuscript describes the mapping of the BACs that supposedly contain retroviral sequences of Jaagsiekte sheep retrovirus (JSRV). Authors found that BACs are localized on homologous chromosome regions in sheep and buffalo. For one BAC there were two locations in the genome. One of the genomic regions had multiple BAC’s localisation. One BAC was localised in the centromere.

It should be indicated whether the authors conducted a PCR experiment to verify whether each BAC clone has an insertion of the enJSRV before conducting FISH.

It has been better clarified now (see material and methods section and table 1)

I have some doubt whether calling mapping of the BACs, containing enJSRV is the same as saying mapping enJSRV, unless it was for a fact shown that these BACs have an insert of the enJSRV AND also shown that genomes of both sheep and buffalo contain enJSRVs. I assume that the authors have purchased the same BACs as in the original paper that has identified 27 enJSRV in the sheep genome. The names of the BACs in the CHORI library have to be listed in the table. However, it happens sometimes that some BACs may not contain the intended sequence. So if authors want to claim that they have mapped not only BACs but BACs containing enJSRV or even claim the mapping of enJSRVs themselves, then we readers have to be sure that enJSRVs are indeed part of these BACs. Otherwise, I would not be calling it mapping of the enJSRV. 

All FISH-mapping data were performed simultaneously (2008-2009) using all 27 BAC we received from our English colleagues. Some data have been used to support some studies performed in the past (and cited in the present MS), but most of the data remained unpublished until now.

Mapping enJSRV itself would include the direct amplification of the enJSRV from a sheep genome, labeling the PCR product, and mapping it onto sheep and buffalo chromosomes. The FISH experiment, of course, would be much trickier in such a case as the probe would be much smaller than BAC. 

As reported above, all BAC containing the enJSRVs have been verified (see material and methods and table 1)

Figure 1. Suggestion - at the bottom left corner of each FISH image put the name of the mapped retrovirus (in the white font?). At the bottom of the image put OAR and BBB under respective ideograms.

Done

The authors say in the discussion that they have previously mapped two type 1 loci into one of the bands that also contains several insertions of enJSRVs. Well, this band has much more genes as shown in Gene Map Viewer or USCIS Genome Browser: for example, genes PRDM5, SYNPO2, NDST3, and many more that are situated between QRFPR and TRAM1L1 genes described in the manuscript. If retrovirus accumulation in this band indeed affects the evolution of the mammalian genome then it makes sense to look at GO functions of ALL the genes in this particular band, not only on the 2 genes mapped by FISH. And then make a conclusion on whether enJSRV inserts may have somehow affected nearby genes.

As requested, a table listing all genes included between QRFPR and TRAM1L1 has been reported in the revised version (see table 3)

It would be interesting to extend this study beyond Bovidae and to find a point on a phylogenetic tree of Cetartiodactyla where the genome integration of the retrovirus has occurred. This can at first be done bioinformatically by mining the cetartiodactyl genomes beyond Bovidae for enJSRVs.

We refer to previous studies reporting these data in both the introduction and discussion.

However, first, we need to be sure that authors are not simply mapping ovine BACs in sheep and goat (those of course would localize on homologous chromosomes in these species!), but mapping BACs, containing enJSRV. Also, we need to be sure that when we are localizing some BAC in buffalo and claiming that the BAC has enJSRV, we are not simply mapping all sequences other than enJSRVs that are present in the BAC but indeed mapping enJSRVs itself.   

These doubts have been clarified in the materials and methods being all BAC-clones well characterized and verified (see also cited paper and table 1))

Some small suggestions are in the pdf text: provide hypotonic treatments details, add the word retrovirus into the title and etc.

Done

Maybe Carlson et al., 2003 did not find homologous sequences in goat for many of the identified enJSRVs in sheep because of the divergence of enJSRVs. This is one potential indication that authors could be not mapping enJSRVs in buffalo but instead simply mapping BACs in buffalo. Meaning, that BAC can still map onto the buffalo chromosomes even if the buffalo genome does not contain enJSRV. 

As reported above all, BAC used have been well characterized and verified.

Also, we need to be sure that the localization of the BAC on two chromosomes (enJSRV-1) is due to the genome containing multiple copies of the enJSRV and not multiple copies of some other genome fragment that is also present in the BAC. All this should either be discussed in the paper or a new probe that includes only enJSRV (produced by cloning the enJSRV into the plasmid or by PCR amplification) should be made and mapped (which would constitute a significant effort: the same or even bigger than the work presented in the manuscript).

We add a sentence in the discussion about it, suggesting that two copies of enJSRV-1 are present in two different chromosomes.

We wish this revised version will find your agreement.

Best regards

Dr. Alessandra Iannuzzi

Round 2

Reviewer 1 Report

The manuscript has been improved. I have just a minor issue. Please add in the results, if it is the case, that you performed FISH  experiments in all individuals collected, and you did no find any polymorphic loci.

Author Response

“If it is the case, that you performed FISH experiments in all individuals collected, and you did not find any polymorphic loci”.

We thank the reviewer for this useful suggestion. In our investigation no other polymorphic enJSRV loci were found in all studied animals of both species, but as reported in the introduction, “we defined polymorphic enJSRVs as ERVs that are found in some but not all individuals of the same species”.

So, it is possible that by investigating an higher number of animals we could find some other polymorphic enJSRV in some animal. However, in the results we better describe this part.

Reviewer 2 Report

Dear Authors,

Thank you for making the required changes in the new version of the manuscript and providing a clear indication that mapped BACs do contain enJSRV sequences.

I will remain with my point of view that a clear sentence stating that BACs have many other genes and sequences in the clone insert and not just enJSRV. These other sequences are getting simultaneously mapped too.

If you include this sentence – then the reader would know that it could be that some of the mapping sites may not contain the enJSRVs.

Only the mapping of the plasmid that has enJSRV as an insert or mapping of the amplified and labeled enJSRV can be a sure indication of the presence of enJSRV in the particular site. Otherwise bioinformatic verification of the presence of the enJSRV in all genome locations mapped in the paper can prove that enJSRV is indeed present at the site of BACs localization.

Table 3 itself is unnecessary and can be moved to the Supplement.

See the point-by-point response in the attached file.

Sincerely,

Polina Perelman

Author Response

You asked to report a clear sentence stating that BACs have many other genes and sequences in the clone insert and not just enJSRV. These other sequences are getting simultaneously mapped too.

We thank the reviewer for this useful suggestion. The reviewer is correct, and we added the following sentence in the discussion of the manuscript: “However, since BACs contain large genomic inserts, it is entirely possible that, beside enJSRVs, we also simultaneously mapped other genes and sequences present on such clones that share homology with some river buffalo chromosomal regions. Indeed, since some BoERV families (i.e., BoERV24, BoERV26, BoERV28, and BoERV29) are specific to Bovinae family [5], the same could be occurred for some enJSRV which are present in Caprinae subfamily (i.e. Ovis aries) but not in Bovinae subfamily (i.e. Bubabus bubalis)”.

  1. In addition, as suggested by referee 2, table 3 has been integrated by adding three genes more (TRNAE-UUC, TRNAS-GGA, TRNAM-CAU) and reported as supplementary materials (table S1). Also, all references of this table have been reported below this table, thus reducing noticeably the number of references reported in the main test (as requested by the editorial staff)

Finally, we would like to point out that, as you and both reviewers requested, we have had the whole manuscript professionally edited.

All authors have read and approved the revised manuscript. We hope that our resubmission is now suitable for inclusion in Animals, and we look forward to hearing from you.

Best regards,

Alessandra Iannuzzi